# Legume Consumption and Blood Pressure Control in Individuals with Type 2 Diabetes and Hypertension: Cross-Sectional Findings from the TOSCA.IT Study

**DOI:** 10.3390/nu15132895

**Published:** 2023-06-26

**Authors:** Marilena Vitale, Annalisa Giosuè, Sabina Sieri, Vittorio Krogh, Elena Massimino, Angela Albarosa Rivellese, Gabriele Riccardi, Olga Vaccaro, Maria Masulli

**Affiliations:** 1Department of Clinical Medicine and Surgery, Federico II University of Naples, 80131 Naples, Italy; marilena.vitale@unina.it (M.V.); annalisa.giosue@unina.it (A.G.); elena.massimino@unina.it (E.M.); rivelles@unina.it (A.A.R.); riccardi@unina.it (G.R.); maria.masulli@unina.it (M.M.); 2Epidemiology and Prevention Unit, Fondazione IRCCS Istituto Nazionale dei Tumori di Milano, 20121 Milan, Italy; sabina.sieri@istitutotumori.mi.it (S.S.); vittorio.krogh@istitutotumori.mi.it (V.K.)

**Keywords:** legume consumption, pulses, type 2 diabetes, hypertension, TOSCA.IT study

## Abstract

Background: Our aims were to evaluate the relationship of habitual legume consumption with blood pressure (BP) control in a large cohort of people with T2D and hypertension, and to investigate whether specific nutritional components of legumes or other foods may contribute to regulate BP levels. Methods: We studied 1897 participants with T2D and hypertension. Dietary habits were assessed through a validated food frequency questionnaire. Sex-specific quartiles of legume consumption were created. Results: Higher legume consumption was associated with a lower intake of energy, carbohydrates, glycaemic load, alcohol, and sodium, and a significantly greater intake of proteins, fat, monounsaturated, polyunsaturated, fibre, potassium, and polyphenols. Significantly lower systolic and diastolic BP values were observed in the highest vs. lowest quartile of legume consumption (132.9 ± 6.7 vs. 137.3 ± 7.0 mmHg, *p* < 0.001; 78.9 ± 4.1 vs. 81.0 ± 4.2 mmHg, *p* = 0.002; respectively), as well as the proportion of people meeting the treatment targets (61.3% vs. 37.4% and 71.3% vs. 52.4%, respectively, *p* < 0.01). This association was independent from other foods whose consumption is associated with the high legume intake. Conclusions: In people with T2D and hypertension, three servings of legumes per week are associated with significantly better BP control. This gives further support to current dietary guidelines in recommending the frequent consumption of legumes, as a “ready-to-use” dietary strategy to achieve optimal BP control.

## 1. Introduction

Hypertension is a major risk factor for coronary artery disease, stroke, heart failure, peripheral vascular disease, and chronic kidney disease [1]. To date, hypertension affects between 16 and 37% of the population globally [2] and it is approximately twice as common in persons with type 2 diabetes [3,4], reaching a prevalence of about 80–90%. 

Data from observational studies show that high blood pressure levels cause a significant increase in the risk of vascular complications in individuals with diabetes and predispose to a higher incidence of chronic kidney disease, ischemic cerebrovascular disease, retinopathy, and cardiovascular mortality [5,6,7,8]. There is solid evidence that blood pressure reduction in people with diabetes largely improves micro- and macrovascular outcomes. Data from the UK Prospective Diabetes Study showed that each 10 mmHg decrease in systolic blood pressure was associated with 12% reduction in risk to develop complications related to diabetes, 11% reduction in the risk of myocardial infarction, and 13% reduction in microvascular complications [9]. Therefore, the effective management of high blood pressure levels is a major target in individuals with diabetes.

Pharmacological treatment is required in the management of most patients, but lifestyle modification, including a better quality of diet, and weight management, are key components to control both blood glucose and blood pressure. Previous studies have shown that the consumption of specific foods, such as fruit and vegetables, dairy products, and whole grains, or the adherence to specific dietary patterns, such as the Mediterranean diet, or the low glycaemic index diet, can improve blood pressure control in diabetes patients [10,11,12]. The Dietary Approaches to Stop Hypertension (DASH) study, conducted more than 20 years ago [13,14], showed that a diet rich in plant-based foods, wholegrains, low-fat dairy products, and low sodium intake, is effective in the prevention and management of hypertension [15]. 

Among plant-based foods, legumes contain various components linked to lower blood pressure, including dietary fibre, bioactive peptides, and flavonoid polyphenols [16,17], but their relationship with blood pressure control in people with diabetes is little investigated and with discordant results [18]. A recent intervention trial of Jenkins et al., has shown a positive effect of legume consumption on systolic blood pressure in subjects with type 2 diabetes following a low-GI, legume rich diet, compared with a high wheat fibre diet [17]. However, the large daily amount of legumes used in this study (1 cup of legumes per day, i.e., 200 g/day) does not allow to extrapolate the results to real life conditions. Furthermore, another clinical trial aimed at determining the effects on cardio-metabolic risk factors of red meat replacement with legumes in people with type 2 diabetes, found no difference in systolic and diastolic blood pressures after 8 weeks of either legume-free diet or legume-based diet although improvements in other markers of metabolic health were observed [19].

Therefore, the aims of the present study were to evaluate the association between habitual legume consumption and systolic and diastolic blood pressure control in a large cohort of people with type 2 diabetes and hypertension, and to investigate whether specific nutritional components of legumes, or other foods associated with a legume rich dietary pattern, may contribute to modulate blood pressure levels in real life conditions.

## 2. Materials and Methods

### 2.1. Participants

We analysed the baseline data from the TOSCA.IT trial (URL: clinicaltrials.gov (accessed on 31 May 2023) NCT00700856), a randomized clinical trial designed to assess the cardiovascular effects of two different hypoglycaemic drugs. The TOSCA.IT study complied with the Declaration of Helsinki and was granted the approval of the Ethics Committee of the coordinating centre and of each participating centre; written informed consent was obtained from all participants. Details on the study protocol, inclusion and exclusion criteria have been published elsewhere [20,21]. Briefly, 3028 people with type 2 diabetes and with serum creatinine < 1.5 mg/dL were recruited in 60 diabetes clinics distributed all over Italy. For the present analysis, we selected 1897 individuals with diagnosis of hypertension and with valid dietary data. Hypertension was defined as use of anti-hypertensive medications and/or systolic blood pressure (SBP) values ≥ 140 mmHg and/or diastolic blood pressure (DBP) values ≥ 90 mmHg [22].

For the purposes of the present analyses, we used data collected at baseline, prior to randomization to the study treatments.

### 2.2. Measurements

Height and weight were measured, and body mass index (BMI) was calculated as kg/m^2^. Waist circumference was measured according to a standard protocol. Subjects were classified as current smokers when they reported having smoked at all over the previous 12 months or more.

Fasting blood sample were analysed in a central laboratory. High-performance liquid chromatography (HPLC) was used to measure HbA1c. Lipid profile (i.e., total and HDL-cholesterol, and triglycerides) were measured by standard methods. The LDL-cholesterol was calculated by the Friedewald equation for people with triglycerides values < 400 mg/dL [23]. The study included patients inadequately controlled with metformin only (stable treatment for the last two months with metformin in monotherapy at 2 g/day) and HbA1c ≥ 7.0% and ≤9.0% [20]. Blood pressure was measured in a seated position after five minutes rest with a mercury sphygmomanometer. Three measurements were taken at 1 min intervals. The average value was considered in the analysis.

### 2.3. Dietary Assessment

Dietary habits during the past 12 months were assessed by the validated EPIC food frequency questionnaire (FFQ) [24] for a total of 248 questions concerning 188 different food items that were classified into 74 predefined food groups. The respondent indicates the absolute frequency of consumption of each item (per day, week, month, or year) and the quantity by selection of pictures showing a small, medium, and large portion size, with additional quantifiers (e.g., “smaller than the small portion” or “between the small and medium portion”) or by selection of a predefined standard portion when no image was available. Incomplete questionnaires and questionnaires with implausible data (i.e., energy intake less than 800 or greater than 5000 kcal/day) were excluded from the analyses.

Through the use of a specifically designed software developed ad hoc by the Epidemiology and Prevention Unit, Fondazione IRCCS, Istituto Nazionale dei Tumori, Milan [25], questionnaire dietary data were converted into frequencies of consumption and average daily quantities and then linked to Italian food tables to obtain estimates of daily intake of macro- and micro-nutrients and energy. Polyphenol intake was assessed using a combination of USDA and Phenol-Explorer databases [26,27,28,29]. Use of drugs was assessed by interview.

### 2.4. Statistical Analysis

Data are given as numbers and percentages, or mean values (M) and standard deviation (SD) for continuous variables, as appropriate.

To correct for the different energy intake between subjects, the intake of legumes was expressed as g/1000 kcal/day. The sex specific quartiles of legume consumption were created to assess the relation of legume consumption with blood pressure, anthropometric variables, glucose control, and plasma lipids.

To compare variables across the quartiles, the analysis of variance with a test for a linear trend was used for the continuous variables, and the χ^2^ test was used to compare proportions. 

Multivariate linear regression analysis was used to estimate the separate and combined association of legumes and other dietary factors with the variables of interest (i.e., systolic, and diastolic blood pressure) and results were expressed as regression coefficients (β). The explored dietary factors were cereals, wholegrains, potatoes, vegetables, fruits, nuts, fish, meat, eggs, dairy products, milk and yogurt, olive oil, animal fat, alcoholic beverages, sweeteners beverages, coffee and tea, and cakes and pastries. Multivariate linear regression analyses were performed to explore the impact of nutrients or bioactive compounds of legumes (i.e., vegetable proteins from food sources, fibre, sodium, potassium, and polyphenols) on systolic and diastolic blood pressure.

The statistical analyses were performed with IBM version 28.0. Statistical tests were two-sided, and *p* values of less than 0.05 were considered to indicate statistical significance.

## 3. Results

The study population consisted of 1079 men (56.9%) and 818 women (43.1%) with a history of type 2 diabetes 8.7 years (±5.7 years) and hypertension of about 7.2 years (±6.4 years). The mean age was 62.8 years (±6.4 years), and BMI 30.8 ± 4.5 Kg/m^2^. Most participants were taking anti-hypertensive medications (95.6%) and 22% of total population had micro-albuminuria (Appendix A).

The mean values of systolic and diastolic blood pressure, together with the percentage of people on target for systolic (<130 mmHg) and diastolic (<80 mmHg) blood pressure across the sex-specific quartiles of legume consumption are reported in Figure 1. Significantly lower levels of systolic and diastolic blood pressure were observed in the highest quartile of legume consumption (Q4) compared to the lowest (Q1) (132.9 ± 6.7 vs. 137.3 ± 7.0 mmHg, *p* < 0.001; 78.9 ± 4.1 vs. 81.0 ± 4.2 mmHg, *p* = 0.002; respectively). Accordingly, the percentage of people on target for systolic and diastolic blood pressure significantly increased across the quartiles of legumes intake (from 37.4% to 61.3% for SBP and from 52.4% to 71.3% for BDP, respectively (*p* < 0.01) (Figure 1).

No significant differences were observed across quartiles regarding age, diabetes duration, BMI, waist circumference, HbA1c, total cholesterol, LDL- and HDL-cholesterol, triglycerides, C-reactive protein, and percentage of people with micro-albuminuria (Appendix A). The same findings were observed after the exclusion of patients with micro-albuminuria.

The specific food pattern associated with legume consumption is shown in Table 1. A higher legume consumption was significantly associated with a greater intake of wholegrain cereals, vegetables, fruits (all types), fish, eggs, and olive oil (*p* < 0.05), and a lower intake of refined cereals, animal fat (i.e., butter, cream, etc.), alcoholic beverages, and cake and pastries (*p* < 0.05). No significant differences were detected for potatoes, nuts, meat, dairy products, milk and yogurt, sweetened beverages, and coffee and tea consumption (Table 1).

As compared to subjects with lower legume intakes (Q1), participants reporting higher legume consumption (Q4) had significantly lower intake of energy, total carbohydrates and starch, glycaemic load, alcohol, and sodium (*p* < 0.001) (Table 2). Significantly greater intake of total proteins, especially from vegetable food sources, total fat, especially monounsaturated and polyunsaturated fat, fibre, potassium, and polyphenols, were also associated with higher legume consumption (*p* < 0.05). The intake of saturated fat, cholesterol and added sugar, and the glycaemic index of the diet did not change across the quartiles (Table 2).

In multivariate analysis, systolic and diastolic blood pressure values were significantly and inversely associated with legume consumption (β = −0.087, *p* < 0.001; β = −0.061, *p* = 0.014, respectively) independently of any other dietary item; none of the other foods consumed by the study participants was associated with blood pressure (Table 3). 

In addition, in order to evaluate which nutrients linked legume consumption to systolic and diastolic blood pressure values, separate linear regression models were used. As reported in Table 4, the only dietary component associated with systolic and diastolic blood pressure values was dietary fibre (β = −0.139, *p* = 0.007; β = −0.111, *p* = 0.031, respectively). This association disappeared when legumes were included in the model and was replaced by a significant inverse association between systolic and diastolic blood pressure values with legume consumption, thus suggesting that dietary fibre plays a significant role in modulating the relationship between legume consumption and blood pressure control.

## 4. Discussion

This study investigated, for the first time, the association between legume consumption and blood pressure in a large cohort of people with type 2 diabetes, in the context of real-life conditions, where the amount of legumes intake was that habitually consumed by the study population and substantially lower than in the intervention trials conducted so far [17,19,30]. Notably, in this cohort the highest quartile of legume consumption corresponded to an intake of 28.2 g/1000 kcal/day that means 70 g (about 1/2 cup) 3 times a week, an intake fully consistent with the current nutritional recommendation for people with diabetes [31,32]. The results clearly showed that higher intakes of legumes are associated with lower levels of systolic and diastolic blood pressure, and with a greater proportion of people meeting the treatment target for blood pressure. Furthermore, multivariate analyses show that this association is independent from other foods whose consumption is associated with a legume rich dietary pattern, such as cereals, wholegrain foods, vegetables, and fruits, and may be, at least in part, due to the high fibre content of legumes.

Legume consumption in relation to blood pressure has been studied in people with or without type 2 diabetes in intervention studies exploring the beneficial effects of high legume intakes. Jenkins et al. [17] reported a beneficial effect of legume consumption on systolic blood pressure among individuals with type 2 who consumed about 1 cup of legumes per day (190 g/day) for 3 months within the context of a low-GI legume rich diet compared to those following a high wheat fibre diet. Another clinical trial aimed at evaluating the effects on cardio-metabolic risk factors of red meat replacement with legumes in people with type 2 diabetes, found no difference in systolic and diastolic blood pressures after 8 weeks of either a legume-free diet or a legume-based diet [19]. In this substitution study, half a cup of cooked legumes (70 g/day) was substituted for one serving of red meat. Moreover, in a meta-analysis of eight isocaloric dietary pulse intervention trials including 554 participants with and without type 2 diabetes or hypertension a significant improvement of blood pressure was achieved by a median consumption of 162 g/day of pulses [30]. 

A feature shared by all these studies is the high amounts of legumes used within dietary interventions which is unfeasible for the long term in real life conditions. Yet, our study confirms the association of legumes with lower systolic and diastolic blood pressure for a lesser amount of consumption (70 g three times a week), which is more feasible and in line with that recommended for the general population and suggested by the dietary guidelines for people with type 2 diabetes.

Prior population studies in non-diabetic people have also reported that an increased legume intake is associated with a reduced risk of developing hypertension. A meta-analysis of five observational studies reported a ~5% risk reduction to develop hypertension for about 70 g/day of legume consumption [33]. A subsequent work on 83,284 incident cases of hypertension confirmed a protective association for the highest compared to the lowest levels of legume intake [34]. The 1999–2002 National Health and Nutrition Examination Survey (NHANES) has shown that US adults who consumed approximately 1/2 cup per day (1 serving) of cooked dry beans or peas have lower odds of elevated blood pressure and a 1.7 mm/Hg lower mean systolic blood pressure than non-consumers [35]. Recent data from the EPIC Norfolk cohort also showed that legume consumption between 55–70 g/day was associated with a lower subsequent risk of hypertension [36].

All these previous observational studies were performed in people without diabetes; conversely, we performed our study in a large cohort of individuals with type 2 diabetes; it is of interest that our population was almost entirely under anti-hypertensive pharmacological treatment. This means that legume consumption can represent an additional tool—besides medications—to lower blood pressure, allowing a larger proportion of people with type 2 diabetes to reach the target. 

Among several legume components potentially involved in the beneficial relationship observed with blood pressure in people with type 2 diabetes, like proteins, fibre, sodium, potassium, and polyphenols, only fibre showed a significant association with lower levels of systolic and diastolic blood pressure according to our results. This can be due, at least in part, to the demonstrated role of fibre in reducing inflammation [37,38] and improving the endothelial function [39]. When the same analysis was performed adding as covariate legume consumption, only legumes were significantly and inversely associated with systolic and diastolic blood pressure, while the relationship with dietary fibre disappeared, thus demonstrating that the high fibre content of legumes plays an important role in the improvement of blood pressure control associated with consumption of this food group. 

However, the possible contribution of other macro- and micro-nutrients present in legumes in explaining the association between consumption of this food group with a better blood pressure control should also be considered. In this respect, the vasodilatory effect of arginine in legume proteins via converting into nitric oxide has been demonstrated in animal and in vitro models and proposed as a potential mechanism driving the relationship observed [40,41]. Other bioactive peptides obtained from legume sources have been shown by in vitro studies to have high antioxidant capacity and the potential to inhibit the dipeptidyl peptidase-IV and the Angiotensin Converting Enzyme [42]. The inhibition of these two enzymes is the therapeutic target of drugs used for type 2 diabetes mellitus and for hypertension, while the antioxidant activity can prevent the development of several chronic diseases related to oxidative stress. However, more studies are needed to understand the bioavailability of these molecules, since the peptides released from beans can be inactivated during further digestion by peptidases in the intestinal epithelial cells.

The health benefits of polyphenols from legumes in relation to blood pressure control were summarized in a recent comprehensive review by Ganesan et al. [43] demonstrating that bioactive components, particularly phenolic compounds, play an important role in modulating vascular integrity and inflammatory markers, two factors associated with improvement of blood pressure.

There are several strengths of this study. This is the first large cohort of people with type 2 diabetes in which the links between legume intake and blood pressure were investigated in real-life conditions. Furthermore, the study was able to control for medications, metabolic parameters, and dietary components, which can play a significant modulating effect on blood pressure. Finally, the use of a central laboratory for biochemical analyses ensures good quality standards for the measurements. 

Our study also has limitations, the most relevant being the cross-sectional study design which does not enable the establishing of a cause–effect relationship. Residual confounding cannot be ruled out, even though our findings are independent of (1) other dietary items and their components, such as mono- and poly-unsaturated fatty acids, glycaemic load, sodium, and alcohol intake; and (2) the main potential confounders, such as age, BMI, and smoking habit, were controlled. Furthermore, we did not analyse the role of different types of legumes consumed, as legume intake was aggregated into total amount. In this regard, we cannot exclude that different types of legumes may have different effects due to their distinct nutritional characteristics. Finally, we cannot exclude that individuals eating a healthier diet and consuming less alcohol may be more adherent to their medication.

## 5. Conclusions

Our findings in people with type 2 diabetes showed that the association of legume consumption with lower systolic and diastolic blood pressure values is already present for an intake of approximately three servings per week, which is feasible and in line with the current dietary recommendations for people with diabetes. This gives further support to current dietary guidelines in recommending the use of legumes (beans, peas, chickpeas, and lentils) for people with type 2 diabetes.

## Figures and Tables

**Figure 1 nutrients-15-02895-f001:**
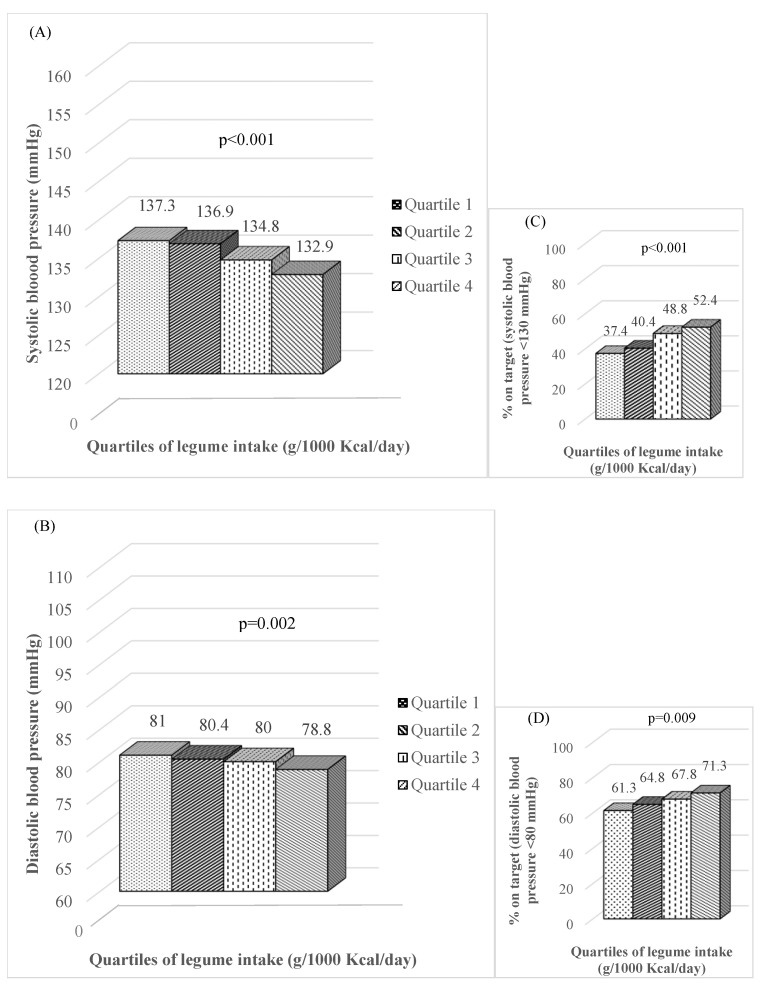
Systolic (**A**) and diastolic (**B**) blood pressure, expressed as absolute values and proportion of people on target (systolic blood pressure < 130 mmHg (**C**), diastolic blood pressure < 80 mmHg (**D**)), according to the sex-specific quartiles of legume intake in people with type 2 diabetes and hypertension.

**Table 1 nutrients-15-02895-t001:** Food intake (g/1000 kcal/day) in people with type 2 diabetes and hypertension according to the sex-specific quartiles of legume intake (g/1000 kcal/day).

	Quartile 1 (2.9 ± 1.7)	Quartile 2 (8.1 ± 1.8)	Quartile 3 (14.1 ± 2.7)	Quartile 4 (28.3 ± 10.6)	*p*-Value
Cereals (pasta, rice, bread)	98.5 ± 40.7	98.1 ± 35.9	93.9 ± 33.9	86.2 ± 33.6	<0.001
Wholegrain cereals (bread)	6.7 ± 15.7	6.5 ± 13.8	6.9 ± 15.6	9.7 ± 18.3	0.006
Potatoes	9.2 ± 12.3	10.6 ± 14.9	11.0 ± 9.3	11.0 ± 12.6	0.074
Vegetables (salad, cooked vegetable)	87.1 ± 55.6	86.7 ± 41.8	98.1 ± 47.7	109.9 ± 49.1	<0.001
Fruits (all type)	148.7 ± 89.4	150.8 ± 88.1	156.9 ± 90.5	185.8 ± 106.9	<0.001
Nuts	0.67 ± 2.12	0.50 ± 0.86	0.51 ± 0.93	0.57 ± 1.43	0.238
Fish	19.2 ± 17.7	21.1 ± 14.7	23.1 ± 16.5	27.9 ± 20.8	<0.001
Meat (red, white, and processed)	53.5 ± 28.8	51.8 ± 23.9	53.0 ± 25.4	53.1 ± 25.2	0.776
Eggs	9.4 ± 7.8	10.2 ± 6.5	10.7 ± 7.0	11.2 ± 7.9	0.002
Dairy Products	20.7 ± 14.8	20.6 ± 13.8	20.0 ± 12.6	19.5 ± 12.7	0.491
Milk and Yogurt	108.4 ± 105.4	104.7 ± 101.6	110.8 ± 100.0	117.0 ± 110.6	0.320
Olive Oil	12.2 ± 6.3	13.0 ± 5.2	14.2 ± 5.5	15.8 ± 6.3	<0.001
Animal fat (butter, cream, etc.)	1.72 ± 1.78	1.60 ± 1.58	1.50 ± 1.46	1.24 ± 1.30	<0.001
Alcoholic beverages (wine, beer)	72.0 ± 98.0	64.1 ± 89.8	61.4 ± 81.2	51.3 ± 75.4	0.003
Sweetener beverages	29.9 ± 64.0	26.5 ± 56.7	24.6 ± 46.9	25.4 ± 48.4	0.466
Coffee and tea	80.8 ± 75.9	73.3 ± 59.1	76.9 ± 55.7	83.2 ± 72.5	0.105
Cake and pastries	20.9 ± 21.6	21.9 ± 18.8	18.3 ± 17.7	15.4 ± 14.3	<0.001

M ± SD.

**Table 2 nutrients-15-02895-t002:** Energy and nutrient composition of the habitual diet in people with type 2 diabetes and hypertension according to the sex-specific quartiles of legume intake (g/1000 kcal/day).

	Quartile 1 (2.9 ± 1.7)	Quartile 2 (8.1 ± 1.8)	Quartile 3 (14.1 ± 2.7)	Quartile 4 (28.3 ± 10.6)	*p*-Value
Energy (kcal/day)	1908 ± 821	1996 ± 730	1929 ± 795	1655 ± 607	<0.001
Proteins (% of TE)	18.1 ± 2.7	18.1 ± 2.4	18.3 ± 2.5	18.6 ± 2.7	0.006
from animal food sources (% of TE)	12.6 ± 3.3	12.5 ± 3.1	12.7 ± 3.1	12.9 ± 3.3	0.452
from vegetable food sources (% of TE)	5.5 ± 1.2	5.6 ± 1.2	5.6 ± 1.1	5.8 ± 1.0	0.006
Total fat (% of TE)	36.0 ± 6.5	36.5 ± 5.6	37.1 ± 5.9	37.7 ± 6.2	0.001
SAFA (% of TE)	12.3 ± 2.7	12.4 ± 2.5	12.3 ± 2.5	12.0 ± 2.5	0.187
MUFA (% of TE)	17.2 ± 4.0	17.6 ± 3.4	18.2 ± 3.6	18.8 ± 4.0	<0.001
PUFA (% of TE)	4.3 ± 1.1	4.3 ± 1.0	4.5 ± 1.2	4.7 ± 1.2	<0.001
Cholesterol (mg/1000 kcal)	177.3 ± 56.2	183.2 ± 50.0	182.5 ± 49.2	184.5 ± 54.3	0.162
Carbohydrates (% of TE)	45.7 ± 8.0	45.3 ± 7.0	44.5 ± 7.2	43.6 ± 7.4	<0.001
Starch (% of TE)	29.3 ± 9.0	29.0 ± 8.1	28.2 ± 7.8	26.4 ± 7.7	<0.001
Soluble carbohydrates (% of TE)	16.4 ± 6.0	16.3 ± 5.6	16.2 ± 5.4	17.2 ± 5.5	0.030
Added Sugar (% of TE)	2.6 ± 3.8	2.4 ± 3.5	2.3 ± 2.7	2.3 ± 3.2	0.263
Fibre (g/1000 kcal)	9.5 ± 2.4	9.9 ± 2.3	10.8 ± 2.4	12.7 ± 2.8	<0.001
Glycaemic Index (n)	51.7 ± 3.6	51.8 ± 3.1	51.6 ± 3.4	51.5 ± 3.6	0.729
Glycaemic Load (%)	117.5 ± 64.0	120.0 ± 51.4	113.8 ± 57.2	98.7 ± 41.6	<0.001
Alcohol (g/day)	12.3 ± 17.3	11.6 ± 18.1	10.4 ± 14.7	7.8 ± 12.6	<0.001
Sodium (mg/day)	2158 ± 1186	2250 ± 1108	2136 ± 1076	1767 ± 851	<0.001
Potassium (mg/day)	2883 ± 1093	3068 ± 1029	3089 ± 1124	2913 ± 1010	0.003
Polyphenols (mg/1000 kcal/day)	345 ± 162	348 ± 139	371 ± 144	417 ± 168	<0.001

M ± SD. TE: Total Energy; SAFA: Saturated Fatty Acids; MUFA: Monounsaturated Fatty Acids; PUFA: Polyunsaturated Fatty Acids.

**Table 3 nutrients-15-02895-t003:** Multivariate linear regression model assessing the relationship of individual foods with systolic and diastolic blood pressure.

	Systolic Blood Pressure		Diastolic Blood Pressure	
	ß-Coefficient	*p*-Value	ß-Coefficient	*p*-Value
Cereals (pasta, rice, bread)	−0.046	0.302	0.017	0.703
Wholegrain cereals (bread)	−0.004	0.863	−0.019	0.422
Potatoes	−0.004	0.853	0.016	0.508
Legumes	−0.087	<0.001	−0.061	0.014
Vegetables (salad, cooked vegetables)	−0.005	0.889	0.050	0.196
Fruits (all type)	−0.023	0.422	−0.003	0.905
Nuts	0.018	0.455	−0.013	0.572
Fish	0.036	0.122	0.025	0.551
Meat (red, white, and processed)	0.060	0.239	0.050	0.330
Eggs	0.010	0.688	0.005	0.844
Dairy products	0.043	0.127	0.016	0.566
Milk and yogurt	0.008	0.247	−0.039	0.242
Olive oil	0.008	0.848	−0.044	0.309
Animal fat (butter, cream, etc.)	0.044	0.079	0.052	0.057
Alcoholic beverages (wine, beer)	0.022	0.477	0.055	0.080
Sweetener beverages	0.024	0.342	0.049	0.062
Coffee and tea	0.027	0.265	−0.001	0.964
Cake and pastries	−0.012	0.733	0.035	0.305

**Table 4 nutrients-15-02895-t004:** Multivariate linear regression model assessing the relationship of nutrients and bioactive compounds with systolic and diastolic blood pressure.

	Systolic Blood Pressure		Diastolic Blood Pressure	
	ß-Coefficient	*p*-Value	ß-Coefficient	*p*-Value
Model 1				
Proteins from vegetable food sources	−0.004	0.946	0.065	0.233
Fibre	−0.139	0.007	−0.111	0.031
Sodium	0.021	0.628	0.017	0.700
Potassium	0.087	0.073	0.060	0.143
Polyphenols	−0.035	0.175	−0.031	0.223
Model 2				
Legumes	−0.089	<0.001	−0.082	0.002
Proteins from vegetable food sources	−0.052	0.357	0.021	0.706
Fibre	−0.059	0.287	−0.038	0.493
Sodium	0.011	0.797	0.008	0.859
Potassium	0.061	0.091	0.037	0.378
Polyphenols	−0.031	0.228	−0.028	0.280

## Data Availability

Not applicable.

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
