# Peer review of "Legume Consumption and Blood Pressure Control in Individuals with Type 2 Diabetes and Hypertension: Cross-Sectional Findings from the TOSCA.IT Study"

_nutrients, 2023, doi:10.3390/nu15132895_

Round 1
Reviewer 1 Report
This cross sectional study investigates an important question. I have a few concerns regarding the findings.
1. The definition of hypertension includes taking antihypertensive medication. Can you define how many of the participants in this study were taking antihypertensives for renal protection rather than for hypertension?
2. There is no discussion of adherence to antihypertensive medication. It is quite possible that individuals eating a healthier diet and consuming less alcohol may be more adherent to their medication. This should be mentioned as a likely confounder.
3. The Hosseinpour- Niazi study only used substitution of 3 meals a week with lentils so not a large amount as implied in line 250. Can you explain why they found a benefit on lipid levels and you have not? And why they found no difference in blood pressure?
4. You have converted total intake of legumes to g/1000kcal per day. This means your highest quartile is actually eating about 45g legumes per day (more than in the Hosseinpour- Niazi study). Supplemental table 2 indicates they are eating 9 fold more legumes than quartile 1! This should be commented on.
5. Can you comment on the disparity between the kcal consumed by quartile 4 without lower bmi, waist circumference or lipids?
Overall the quality of English is very good
Minor english corrections- line 37 typo form instead of from
line 50 vegetables
line 64 extension of the results
line 67 However it found benefits in other significant markers of metabolic health
Reviewer 2 Report
This is an interesting and well-written study exploring legume consumption and blood pressure control in individuals with type 2 diabetes and hypertension. However, I have some remarks:
1) In this study hypertension was defined as the use of antihypertensive medications and /or systolic blood pressure (SBP) values ≥140 mmHg and/or diastolic blood pressure (DBP) values ≥90 mmHg. However, in those patients with high cardiovascular risk and/or chronic kidney disease blood pressure below 130/80 mmHg should be targeted.
2) There is no information about renal function and the percentage of patients with chronic kidney disease which is a significant factor for unregulated hypertension. Please add that data in supplementary table 1 and 2.
3) There is no information about antihypertensive therapy.
4) All patients with diabetes were on metformin although they have a long duration of diabetes (over 8 years) and increased HbA1c. Discuss that.
5) Blood pressure was measured in a seated position after five minutes of rest. With a digital sphygmomanometer? Blood pressure was measured only one time or twice?
6) It is well known that the most common side effects reported by people who take metformin are nausea and diarrhea. Whether patients with a higher intake of legumes had a higher risk of metformin side effects?
Round 2
Reviewer 1 Report
Thankyou for your responses to the reviewer questions. I think the comment re legumes being associated with a reduction in blood pressure independent of drugs in the abstract needs to be modified and "independent of drugs" should be removed given the likelihood that those who had a healthier diet may be more adherent to their medication which was what resulted in the lower blood pressure rather than the legume intake(as well as the added comment in the limitations of the study)
Author Response
We agree with the reviewer. We have deleted the words “independent of drugs” from the abstract.
Reviewer 2 Report
In your response letter, you indicated that patients with serum creatinine >1.5 mg/dl were excluded by protocol (Page 2, Lines 83-84). However, serum creatinine is not an optimal measure of kidney function particularly in patients with diabetes because of obesity (in this study BMI is around 30 kg/m2). It is an estimated glomerular filtration rate below 60 mL/min/1.73m2 which corresponds to serum creatinine over 1.5 mg/dL used in this study.
Please change "By protocol, all patients were taking the same treatment for diabetes (i.e., metformin 2 g/day) [20]" to "Study included patients inadequately controlled with metformin only (stable treatment for the last two months with metformin in monotherapy at 2 g/day) and HbA1c ≥ 7.0% and ≤ 9.0% [20]"
Author Response
The sentence has been modified according to the reviewer suggestion (Page 3, Lines 102-104).